

# Reciproc vs. hand instrumentation in dental practice: a study in routine care

Andreas Bartols[1,2], Claudius A. Reutter[3], Bernt-Peter Robra[4] and Winfried Walther[1]

[1] Dental Academy for Continuing Professional Development, Karlsruhe, Germany
[2] Clinic for Conservative Dentistry and Periodontology, Christian-Albrechts-University Kiel, Kiel, Germany
[3] Private Dental Practice, Bellheim, Germany
[4] Institute of Social Medicine and Health Economics, University of Magdeburg, Magdeburg, Germany

Corresponding author
Andreas Bartols,
andreas_bartols@azfk.de

## ABSTRACT

**Background.** Little is known about the clinical impact of new root canal preparation systems in general dental practice under routine care conditions. Therefore, we compared hand instrumentation (H) with Reciproc (R) (VDW, Munich, Germany) preparation. The outcomes were endodontic related pain and oral health related quality of life (OHRQoL), evaluation of the procedures by the patients and the strain felt by the dentists during root canal therapy.

**Methods.** Six dentists participated in the trial as practitioner–investigators. In the first phase of the trial they prepared root canals with H and in the second phase with R. The patients documented their pain felt with a visual analogue scale (VAS 100) and OHRQoL with the German short version of the oral health impact profile (OHIP-G-14) before treatment and before the completion of therapy and answered questions about how they experienced the treatment. The dentists documented their physical strain during treatment.

**Results.** A total of 137 patients were included in the evaluation. 66 patients were treated with H, 71 with R. Pain reduction was 32.6 (SD 32.9) VAS (H) vs. 29.4 (SD 26.9) VAS (R) ($p = 0.550$), and the improvement of the OHIP-14 score was 5.5 (SD 9.2) (H) vs. 6.7 (SD 7.4) (R) ($p = 0.383$). There were no statistical differences in both groups. Significantly fewer patients felt stressed by the duration of treatment with R as with H ($p = 0.018$). Significantly more dentists reported that their general physical strain and the strain on their fingers were less severe with R than with H ($p = 0.013$ and $p < 0.001$).

**Discussion.** H as well as R effectively reduced endodontic related pain and OHRQoL without statistical differences. R has advantages in terms of how patients experience the treatment and regarding the physical strain felt by the dentists.

## INTRODUCTION

There is a wide selection of instruments for root canal preparation (*Hargreaves, Cohen & Berman, 2011*). Whereas clinical data are available for hand instruments and rotary root

canal preparation systems (*Fleming et al., 2010*), we know little about the impact in general dental practice under routine care conditions when the newest single-file systems are used.

In 2008, Yared was the first one to describe the preparation of root canals using only one instrument with reciprocating motion (*Yared, 2008*). The further development of this concept led to the introduction of the Reciproc System (VDW, Munich, Germany), but also to other reciprocating instrument systems as WaveOne (Dentsply, Konstanz, Germany), Genius files (Ultradent, South Jordan, UT, USA) or the Twisted Files Adaptive System (Kerr, Orange, CA, USA) with a combination of rotary and reciprocating movement. Besides reducing the number of instruments to one, Reciproc files are currently the only instruments that were also intended to allow root canal preparation without glidepath preparation (*Yared, 2011*), while all other above mentioned systems were not primarily designed for this purpose. *In vitro* studies prove that mechanical root canal preparation systems are superior to hand instrumentation in relation to various parameters, such as root canal straightening, preparation faults and preparation time (*Kumar et al., 2013*; *Schäfer & Florek, 2003*). The effectiveness gap between *in vitro* studies and clinical reality in general dental practice, if any, is unknown. Therefore, our objective was to examine the effects of the new instruments under heterogeneous conditions of everyday life of different general dental practices. This goal requires a simplified pragmatic approach.

For the patients it is most important to eliminate endodontic related discomfort and to provide a comfortable treatment experience. Root canal therapy (RCT) is a common and effective treatment for the relief of endodontic-related pain (*Pak & White, 2011*). Whether there are any differences in effectiveness between hand instrumentation and Reciproc preparation has not been investigated so far. Moreover, there are no studies to the knowledge of the authors on whether patients consider different instrument systems more agreeable (or disagreeable) than others so that their experience differs during treatment. Another dimension rarely taken into account is the physical strain to which the dentist is exposed when using different methods of preparation under general dental practice conditions.

This is why research is needed in the field of new endodontic instruments. For the purpose of such research, we formed a network of several dental practices and performed a multicenter trial (*Patsopoulos, 2011*). For our trial we investigated processes of dental care under everyday circumstances and on patient-relevant outcomes (*Pfaff, Nellessen-Martens & Scriba, 2011*).

The following two-sided hypotheses were investigated in the study:

Does root canal preparation using Reciproc lead to more or less reduction of patients' endodontic related pain and oral health related quality of life (OHRQoL) compared to using hand instruments?

Do patients feel the Reciproc preparation to be more or less comfortable compared to hand instrumentation?

Can more root canals be prepared without glidepath preparation with Reciproc compared to hand instrumentation?

Is the physical strain on the GDP during root canal preparation with Reciproc lower compared to hand instruments?

## METHODS

We conducted the present study as a clinical multicenter trial under routine care conditions. The study involved nine dentists as practitioner investigators (PIs). They recruited a sample of consecutive patients who needed a root canal treatment. The two phases of the study were separated by the training period for the new treatment method. The authors of this study did not act as PIs, but solely as investigators.

In the first study phase endodontic therapy was performed in the usual way with hand instruments. The PIs consecutively included all patients in the study who required endodontic therapy. Then the PIs were trained for the use of the Reciproc System (VDW, Munich, Germany). The one-day training course explained the theoretical bases of the Reciproc System (VDW, Munich, Germany) for root canal preparation and included hands-on training on extracted teeth. At the end of the training course every participating dentist was technically able to prepare root canals by the new method in a reliable way. The training period was short on purpose because we wanted to simulate a common situation of changing treatment methods in general dental practice regarding the timeline sequence.

In this way we calibrated the investigators and reduced the organizational demands on the dental offices, because they did not have to perform two different methods of treatment at the same time.

In the second phase of the study the PIs treated their patients with the Reciproc instruments (R) exclusively. Each of the two treatment phases lasted for 2–3 months.

The study was made in conformity with the Declaration of Helsinki and the Professional Code for Physicians of the Medical Council of the State of Baden-Württemberg. The PIs informed every potential participant personally about the purpose and extent of the study and about the data protection aspects (pseudonymization of data, erasure of the data at the moment of withdrawal from the study). Before the patients were enrolled, they had to submit signed informed consent to the study and to data storage. The Institutional Review Board of the Baden-Württemberg Medical Council reviewed the study and approved it (AZ: F-2011-081-z).

### Patient eligibility and recruitment

The structural similarity of the patients in the two phases was enhanced by inclusion and exclusion criteria. The following criteria were defined for patients' enrollment in the study: the patient had to be at least 18 years old and in need of initial orthograde root canal treatment of one tooth. The exclusion criteria were: hopeless teeth for periodontal or restorative reasons, patients treated for pain only, dentitions in general need of rehabilitation, several symptomatic teeth requiring endodontic treatment, other oral findings causing pain, craniomandibular dysfunction, and communication difficulties.

### Practitioner investigators (PIs)

The nine recruited dentists were general dental practitioners without endodontic specialization. They all used standard hand instruments for root canal therapy (RCT) and had no routine experience with rotary mechanical instrument systems. In addition, none of them had previously used single-file systems. All dentists followed the ''Good

Clinical Practice: Root Canal Treatment" Guideline of DGZMK (German Society of Dental, Oral and Craniomandibular Sciences) (*Hülsmann & Schäfer, 2005*) which contains essential key points of the Quality guidelines for endodontic treatment of the *European Society of Endodontology* (*2006*).

## Endodontic treatment protocol

The endodontic access cavity was prepared under local anesthesia. Then the tooth was isolated with a rubberdam. In the first phase of the study, depending on the situation, the root canals were explored with stainless steel K-files of ISO sizes 06, 08, 10 and 15, in order to create a glidepath up to ISO 15. Subsequently the root canals were fully prepared with stainless steel K-files by the balanced-force technique when hand instrumentation was used (*Roane, Sabala & Duncanson Jr, 1985*). The working length was determined electrometrically and/or by X-ray. In the second phase of the study the root canals were prepared by means of the Reciproc instruments according to the manufacturer's detailed instructions for use (*Yared, 2011*). Glidepath preparation was not performed and the canal was immediately instrumented with a Reciproc instrument. In all cases the Reciproc Gold Motor was used for root canal preparation. If the PI considered it necessary, he prepared the glidepath. If the dentists wanted a final preparation size that is not included in the Reciproc System, the Reciproc file was followed up by a single hand instrument of the ISO size 35 or 40. After preparation, root canals were irrigated with NaOCl 1–3% and a calcium hydroxide dressing was placed in the root canals or the root canals were filled definitely. In case of a calcium hydroxide dressing the tooth was provisionally sealed with a bacteria-proof filling. The treatment was completed in a subsequent appointment either by root filling or in case that the root canal filling was done in an earlier session, by placing a coronal bacteria-proof definite filling.

## Measuring the reduction of patients' pain and OHRQoL

The endodontic related pain and OHRQoL was measured by a patient questionnaire. Pain was evaluated by the Visual Analog Scale (VAS 100) (*Turk, 2011*) and OHRQoL by the German short version of the oral health impact profile (OHIP-G-14) (*John, Micheelis & Biffar, 2004*) which is the German translation of OHIP-14 (*Slade, 1997*). The patients were asked about the biggest complaints in the week before treatment and in the week before completion of treatment. This was 14 days after initial treatment and in connection with either the placement of the root canal filling or the coronal filling of the tooth. Questionnaires were filled in by the patients before the treatment session started or while the local anesthesia was taking effect.

## Assessment of patients' experience during treatment

At completion of treatment the PI asked four closed questions in a five-stage ordinal scale and one open question about how the patients had experienced the treatment. They asked if the patient was stressed/relaxed during treatment, if the anesthesia was sufficiently deep, if the duration of treatment and the required wide mouth opening had put the patient under stress. The answers covered five categories from completely true to not true at all. One open question was about the most uncomfortable experience patients had during treatment.

**Table 1** Number of patients treated with the different systems by practitioner–investigators.

| PI ID | Studygroup | | Total (N) |
| --- | --- | --- | --- |
| | Hand preparation (N) | Reciproc (N) | |
| 1 | 9 | 12 | 21 |
| 2 | 10 | 12 | 22 |
| 3 | 20 | 14 | 34 |
| 4 | 8 | 9 | 17 |
| 5 | 14 | 18 | 32 |
| 6 | 5 | 6 | 11 |
| Total (N) | 66 | 71 | 137 |

## Assessment of dentist-related physical parameters during treatment

In order to evaluate the physical strain during root canal preparation, the PIs answered two closed questions in a five-stage ordinal scale from "completely true" to "not true at all" after treatment. The PIs were asked if their general physical strain during root canal treatment was high and if after treatment they experienced the strain on their fingers. In addition, they were asked if the treatment was performed without glidepath preparation. The number of required therapy appointments was also recorded.

## Statistics

The data were analyzed with the SPSS (Version 21, Win x64) statistics system.

## RESULTS

Six of the nine recruited PIs presented results suitable for evaluation. The remaining three PIs were excluded from evaluation. In all three cases the PIs could not cope with the organizational demands and the resulting recruitment problems.

In the present study six PIs performed and completed 137 endodontic treatments (Table 1). The PIs treated 66 cases (48.2%) in the first trial phase with hand instruments and 71 cases (51.8%) in the second trial phase with Reciproc instruments. 62 (45.9%) patients were women and 73 (54.1%) were men. The gender distribution did not significantly differ in the two trial phases (Fisher's exact test $p = 0.301$). The PIs treated between 5 and 20 cases per study group (Table 1). The distribution of the treatment cases to the PIs and the study groups was not statistically different ($\chi^2$ 2.14 with 5 DF, $p = 0.830$). The average age of the participants was 52.1 (SD 16.33) years and did not differ in both study phases 52.1 (SD 16.4) vs. 52.0 (SD 16.4).

The return rate of the different questionnaires that were completed by the patients and the PIs was 95.8% for the patients' follow-up pain questionnaire and 100% for the PIs' questionnaires. The return rate of the patients' pain questionnaire before treatment was 97.8% and for the demographic data questionnaire it was 98.5%. In all dental practices some patients did not participate in the study. Their number and characteristics could not be collected. However, it is unlikely that this drop-out in the two study phases was structurally different.

**Table 2** Comparison of pain and oral health related quality of life (OHRQoL) by study group (ANOVA).

| | Hand preparation mean (SD) | Reciproc mean (SD) | Sign. |
|---|---|---|---|
| Pain before treatment (VAS 100) | 43.6 (SD 30.7) | 41.2 (SD 27.7) | (n.s.) |
| Pain before end of treatment | 9.5 (SD 16.5) | 11.5 (SD 18.5) | (n.s.) |
| Pain reduction (VAS 100) | 32.6 (SD 32.9) | 29.4 (SD 26.9) | $p = 0.550$ (n.s.) |
| (Pain reduction)/(SD before treatment) | 1.06 | 1.06 | |
| OHIP-G-14 Score before treatment | 9.2 (SD 9.6) | 10.4 (SD 9.6) | (n.s.) |
| OHIP-G-14 Score before end of treatment | 3.4 (SD 5.4) | 3.5 (SD 6.1) | (n.s.) |
| Improvement of OHRQoL (OHIP-G14) | 5.5 (SD 9.2) | 6.7 (SD 7.4) | $p = 0.383$ (n.s.) |
| (Improvement OHIP-G-14)/(SD before treatment) | 0.57 | 0.70 | |

## Evaluating the reduction of patients' complaints

For the evaluation of the primary outcome criterion, we measured pain reduction by VAS (100) and the improvement of OHRQoL (OHIP-G14 score), each being measured before root canal treatment and before completion of treatment, and compared the two study groups (Table 2).

The mean pain score before root canal treatment with hand instruments (H) was 43.6 (SD 30.7) VAS and with Reciproc (R) it was 41.2 (SD 27.7) VAS and decreased to 9.5 (SD 16.5) VAS (H) and 11.5 (SD 18.5) VAS (R).

The mean OHIP-G-14 score before root canal treatment with (H) was 9.2 (SD 9.6) and with (R) it was 10.4 (SD 9.6) and decreased to 3.4 (SD 5.4) (H) and 3.5 (SD 6.1) (R).

For pain reduction and OHRQoL univariate analysis methods (ANOVA) showed a very similar improvement in both study groups H and R (Table 2). The difference in mean values before and after treatment with respect to the initial standard deviation is in pain greater than in OHRQoL. There was no statistically significant difference between the treatment methods regarding pain reduction and improvement of OHRQoL. Multivariate analysis of variance (MANOVA) taking the additional factor "single vs. multiple-visit treatment" into account did not reveal any significant influence of the factors "study group (H or R)," "single vs. multiple-visit" or an interaction of the two regarding both pain reduction and improvement of OHRQoL (all $p > 0.250$).

## Evaluating patients' experience during treatment

Most patients did not experience the root canal therapy as uncomfortable, regardless of the method used. Most patients were rather relaxed during treatment, considered the depth of anesthesia to be adequate and felt no great stress due to wide mouth opening (Fig. 1). The two methods of root canal preparation did not differ in any of the three questions. In terms of duration of treatment, however, the patients experienced the treatment with Reciproc as significantly less stressful (Mann–Whitney-$U$ Test $p = 0.018$).

The most frequent answer to the open question of what patients felt to be most uncomfortable during treatment was "nothing" in both study groups. In the ranking of answers given, more patients said that the root canal preparation itself was more

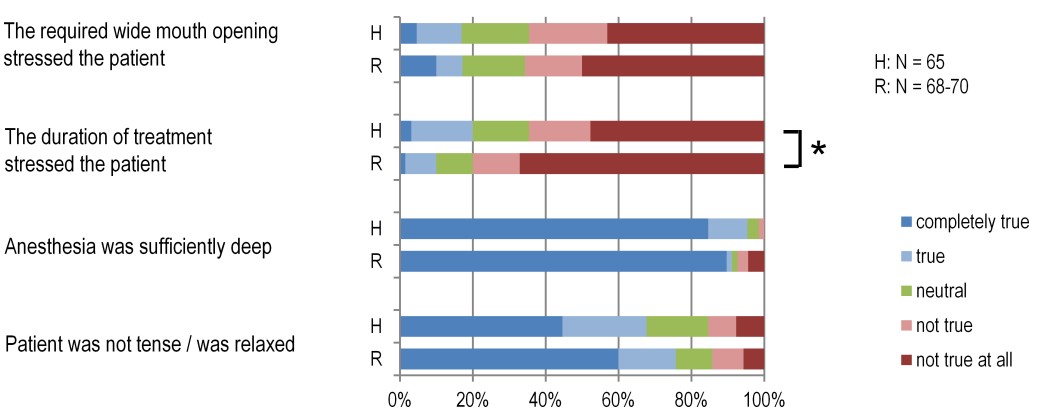

**Figure 1 Patients' feelings during treatment in treatment groups.** In the Reciproc group (R) patients felt significantly less stress during treatment than in the hand preparation group (H) (*Mann–Whitney-U Test $p = 0.018$).

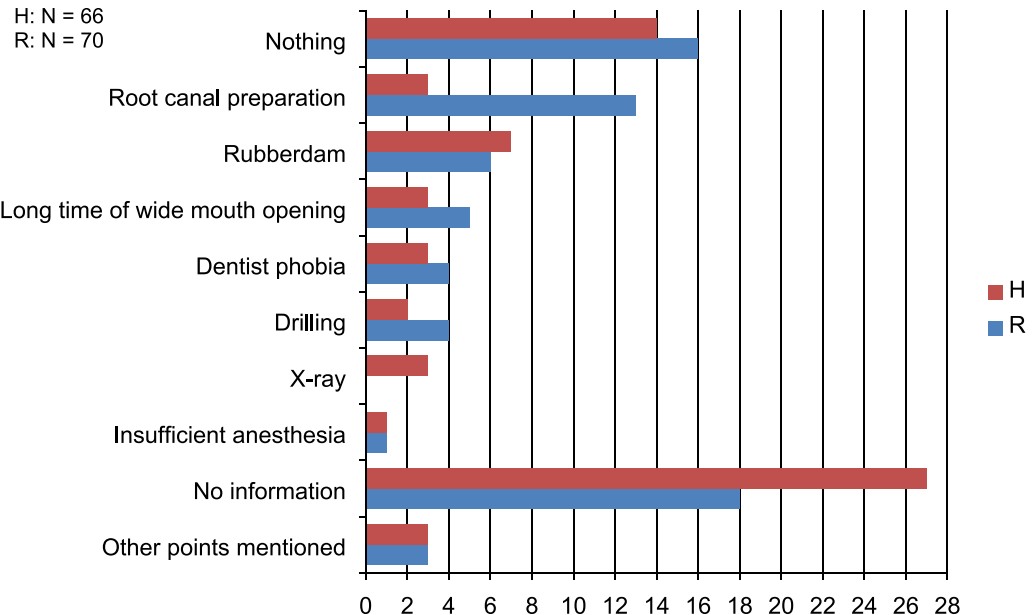

**Figure 2 Patients' answers to the open question what was most uncomfortable during treatment (Number of answers ($N$)).**

uncomfortable with Reciproc, whereas few patients said this about hand instrumentation. In addition, some patients considered the rubber dam uncomfortable (Fig. 2).

## Assessment of dentist-related physical parameters during treatment

The PIs were significantly less strained with the Reciproc System than with hand instrumentation concerning both general physical strain and the strain on their fingers (Mann–Whitney-U Test $p = 0.013$ and $p < 0.001$) (Fig. 3).

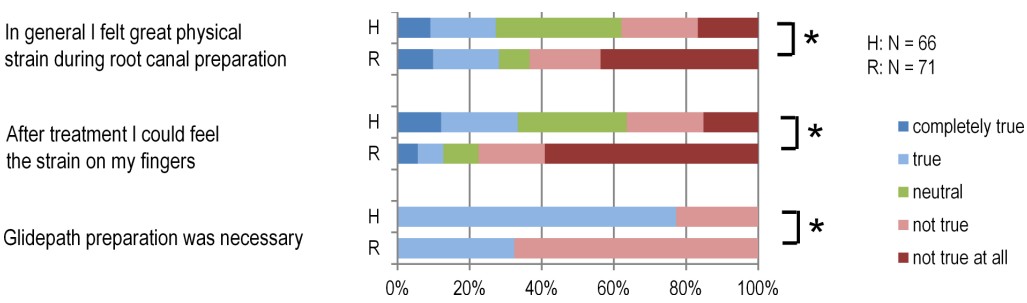

**Figure 3** **Physical strain on dentists.** In the Reciproc group (R) the PIs felt significantly less physical and finger strain than in the hand preparation group (H) (Mann–Whitney-U Test $p = 0.013$ and $p < 0.001$). In the Reciproc group glidepath preparation was significantly less often required (Mann–Whitney-U Test $p < 0.001$). (* Difference between the groups $p < 0.05$).

Moreover, the PIs treated significantly more root canals without glidepath preparation when they used the Reciproc System (67.6%) than they did with hand instruments (22.7%) ($\chi^2$ 27.7 with 1 DF, $p < 0.001$).

## Number of therapeutic appointments

Basically the number of therapeutic appointments did not differ between the systems. Two PIs, however, treated their patients in three appointments as a matter of principle. If these are excluded from the evaluation, the number of appointments tends to decline with Reciproc (Mann–Whitney-$U$ Test $p = 0.035$). With hand instrumentation 34 cases were completely prepared in one appointment, 12 in two appointments and 1 in three appointments, whereas with Reciproc 42 cases were prepared in one appointment, 5 in two and none in three appointments.

## DISCUSSION

Hand instrumentation as well as Reciproc preparation of root canals effectively reduced endodontic related pain and OHRQoL. As the preparation with Reciproc has more advantages in terms of patient comfort and dentist comfort, Reciproc is the technique to be preferred in this respect.

### Measuring the reduction of patients' discomfort

The discomfort indicating the need for endodontic treatment was measured in our study by a combination of the Visual Analog Scale for pain and OHIP-G-14. Thus, the evaluation of the therapeutic procedures is based on measuring pain and the oral-health-related quality of life. The VAS is simple and has good evidence of construct validity (*Turk*, *2011*). It is widely referred to in the endodontic literature (*King et al.*, *2012*; *Martin-Gonzalez et al.*, *2012*; *Pak*, *2012*; *Udoye & Jafarzadeh*, *2011*). The large number of scores (100) makes VAS more sensitive to changes in the intensity of pain felt as compared to scales with fewer answer categories (*Turk*, *2011*). Besides pain felt, the quality of life is an important patient-related outcome (*Pfaff, Nellessen-Martens & Scriba*, *2011*). The only validated measuring instrument for oral-health-related quality of life in German language is OHIP-G (*John et al.*, *2003*). To limit the questionnaire to a length that can still be handled,

we used OHIP-G-14 in our study (*John, Micheelis & Biffar*, *2004*). At the time when our trial was planned in 2011, OHIP was used rarely for endodontic studies (*Dugas et al.*, *2002*; *Gatten et al.*, *2011*).

## Results

In the present study, we compared two basically different methods of root canal preparation. In the first study phase the PIs performed the endodontic therapy in their dental offices as usual with hand instruments. After training for the use of the Reciproc instruments, the PIs prepared the root canals with the new system in the second phase of the study.

The introduction of new methods of treatment always raises the question about their benefit. The new system should have the same or better clinical results than the previous ones and offer an additional benefit at the same time. We defined the reduction of endodontic related discomfort, i.e., reduction of pain and improvement of oral-health-related quality of life, as the most important short-term patient-relevant outcome.

In pain assessment we found a reduction of the mean pain intensities before treatment by 44.6 (H) and 41.2 (R) VAS 100 to 9.5 (H) and 11.5 (R) with rather high standard deviations. This agrees quite well with other studies that reported pain reduction after endodontic therapy (*Ehrmann, Messer & Adams*, *2003*; *Pak*, *2012*). The pain measurements made by *Ehrmann, Messer & Adams* (*2003*) resulted in mean levels of 44.4 VAS (SD 26.9) before therapy and declined to 7.5 (SD 15.5) VAS. The mean pain reduction of 36.9 (SD 29.0) was very similar to the levels we found in our study. *Pak & White* (*2011*) in their review also reported that pain intensity decreased from a mean level of 54 (SD 24) before therapy to levels below 10 VAS in the course of one week.

The oral-health-related quality of life improved in our trial from mean OHIP-G-14-scores of 9.2 (H) and 10.4 (R) to 3.4 (H) and 3.5 (R). *Liu, McGrath & Cheung* (*2014*) in their study found mean OHIP-14-scores of 15.4 (SD 10.5) before endodontic therapy. These scores are somewhat higher and difficult to interpret because so far few studies have investigated the correlation between endodontic related pain and oral-health-related quality of life.

In general, the endodontic literature proves that pain is reduced by endodontic therapy (*Ehrmann, Messer & Adams*, *2003*; *Genet, Wesselink & Thoden van Velzen*, *1986*; *Pak & White*, *2011*). However, comparative studies have so far been dealing mainly with different pain medications (*Attar et al.*, *2008*), different dressings (*Ehrmann, Messer & Adams*, *2003*), effects of analgesia (*Ryan et al.*, *2008*) and the difference between single-visit vs. multiple-visit treatments (*Prashanth et al.*, *2011*). Regarding single- versus multiple-visit treatment no differences were found for one week postoperative pain levels (*Figini et al.*, *2008*; *Prashanth et al.*, *2011*). This agrees with our observations, as we did not find statistically significant differences in the MANOVA for single- versus multiple-visit treatments.

There are very few clinical studies, made with low case numbers ($N = 30$ per experimental group), that have investigated pain after root canal preparation with different instrumentation techniques (*Gambarini et al.*, *2013*). In a recent randomized controlled trial, no differences were found in postoperative pain after preparation of root canals with

different rotary or reciprocating canal preparation systems (*Kherlakian et al.*, *2016*), which agrees with our results.

As the main outcome parameter of our study is the post treatment pain reduction, general features of the used instruments (H and R) have to be considered regarding their influence on post treatment pain. All endodontic instruments produce apically extruded debris (*Al-Omari & Dummer*, *1995*; *Ghivari et al.*, *2011*; *Reddy & Hicks*, *1998*) that could lead to an irritation of the periodontal ligament triggering a neurogenic inflammation response (*Caviedes-Bucheli et al.*, *2013*) and following postoperative symptomatic apical periodontitis. Filing motion techniques with hand preparation tend to extrude more debris beyond the apex than rotating motion techniques with hand instruments (*Al-Omari & Dummer*, *1995*) and rotary NiTi preparation techniques (*Reddy & Hicks*, *1998*). A recent review concludes, that reciprocating as well as rotary-file systems produce apical extrusion of debris and expression of neuropeptides in the periodontal ligament (*Caviedes-Bucheli et al.*, *2016*), and that the inflammatory reaction is influenced by the type of movement and file design of the instruments. Reciproc instruments work in an asymmetrical reciprocating movement with a counterclockwise cutting action and a following clockwise releasing motion. In combination with a S-shaped cross-sectional design with relatively much space for debris, helicoidal angles and an instrument tip these instruments are designed to avoid extrusion of debris beyond the apex and remove material out of the root canal effectively (*Caviedes-Bucheli et al.*, *2013*). Consequently, Reciproc instruments showed only slightly more expression of inflammatory peptides in the periodontal ligament after root canal treatment compared to baseline teeth without instrumentation, while hand instrumentation in a filing motion showed the highest rates of expression (*Caviedes-Bucheli et al.*, *2013*). These results must be carefully interpreted in connection with our study because the direct relationship between expressed inflammatory peptides and perceived pain cannot be exactly quantified by now. Theoretically hand instruments should have a higher tendency to postoperative pain compared to Reciproc instruments, but we found no differences in postoperative pain regardless of the instruments used. Perhaps this can be explained by the fact that in our study hand instruments were used in a rotating motion.

In our trial most patients did not consider the endodontic treatment uncomfortable, but a significantly lower number of patients felt stressed by the duration of treatment when Reciproc was used as compared to hand instruments. We interpret this to be the result of the shorter time required for canal preparation which has already been shown in *in vitro* comparisons with multiple-file rotary preparation systems (*Bürklein et al.*, *2012*). However, more patients felt the root canal preparation itself with Reciproc to be more uncomfortable than the preparation with hand instruments. We assume that this is due to the typical cracking sounds of the Reciproc instruments during root canal preparation.

The PIs considered root canal preparation with the Reciproc System significantly less stressful in terms of general physical burden and strain on their fingers than hand preparation. There are few studies dealing with the physical strain on the dentist during root canal preparation. *Onety et al.* (*2014*) found that the muscle groups used in manual root canal preparation differ from those used in mechanical preparation. The study was made in a simulated treatment situation and does not contain any information about the

strain the dentists themselves felt when using the different methods. In general, they are under great physical strain during RCT which is demonstrated by the large number of musculoskeletal disorders among endodontists (*Zarra & Lambrianidis*, *2014*).

The Reciproc System has been designed by the manufacturer for root canal preparation without a glidepath. In our trial the PIs were able to prepare the root canals without glidepath preparation in the majority of cases (67.6%). *De-Deus et al.* (*2013*) in their *in vitro* study found that 93.4% of the root canals could be prepared without glidepath preparation. We believe that the lower rate in our study is due to the fact that the PIs had no experience in either rotary or Reciproc canal preparation before the trial. Since the system was implemented by the PIs after a single training session without the chance of going through a prolonged learning curve, we think the rate obtained is very good.

In our trial the number of appointments required for root canal treatment tended to go down. As far as the authors know, no study has so far been made on the influence of the preparation methods on the number of therapeutic appointments. In general, the treatment result hardly shows whether an endodontic therapy was performed as single-visit or multiple-visit procedure (*Prashanth et al.*, *2011*; *Su, Wang & Ye*, *2011*). A reduction of the required number of appointments is a desirable outcome regarding patient comfort, but does not necessarily reflect long-term biological objectives.

Hand instrumentation and Reciproc preparation effectively reduce endodontic related discomfort. Reciproc, however, has advantages in terms of how the patients feel during treatment and with regard to the physical strain on the dentist. For the patients the duration of treatment with Reciproc is less stressful and the number of appointments tends to be lower. The dentists consider Reciproc to be significantly less physically stressful than hand instrumentation.

## ACKNOWLEDGEMENTS

The authors thank all participating practitioner investigators for their valuable work.

### Funding
The authors received no funding for this work.

### Competing Interests
The authors declare there are no competing interests. CA Reutter works in private dental practice in Bellheim, Germany and declares no competing interests. VDW made Reciproc Gold Motors and Reciproc files available free of cost for the purpose of the trial. There is no conflict of interest between the authors and VDW.

### Author Contributions
- Andreas Bartols and Claudius A. Reutter conceived and designed the experiments, performed the experiments, analyzed the data, contributed reagents/materials/analysis tools, wrote the paper, prepared figures and/or tables, reviewed drafts of the paper.

- Bernt-Peter Robra conceived and designed the experiments, analyzed the data, contributed reagents/materials/analysis tools, wrote the paper, prepared figures and/or tables, reviewed drafts of the paper.
- Winfried Walther conceived and designed the experiments, contributed reagents/materials/analysis tools, wrote the paper, reviewed drafts of the paper.

### Human Ethics

The following information was supplied relating to ethical approvals (i.e., approving body and any reference numbers):

The Institutional Review Board of the Baden-Württemberg Medical Council reviewed the study and approved it (AZ: F-2011-081-z).

### Data Availability

The raw data has been supplied as Data S1.

### Supplemental Information

Supplemental information for this article can be found online at http://dx.doi.org/10.7717/peerj.2182#supplemental-information.

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
