# Peer review of "Reciproc vs. hand instrumentation in dental practice: a study in routine care"

_PeerJ, doi:10.7717/peerj.2182_

## Round 0.1 · original submission · Minor Revisions

Please respond to all of the comments of the two reviewers.

·

Basic reporting

basic reporting is OK, well written paper

Experimental design

Generally the design is OK. There is some concern on the moment that the questionnaires were presented to the patients: in the week before completion of the treatment. for one-session endo's in that case the root canals were filled, while for the multivisit endo's the filling has to be placed still. This is a variable in the protocol which has to be taken into account

Validity of the findings

results are well formulated and statistically sound (see previous remark)

Additional comments

if the variation between single and multisession endo's is included in the analysis of the questionnaires, the paper is OK for placement

·

Basic reporting

I believe the article meets Peer J standards. There are a number of relatively minor spelling and grammatical errors which I can provide suggested corrections for. It would have been appropriate to at least mention other reciprocating instrument systems currently on the market (eg, WaveOne <Dentsply> and Genius Files <Ultradent>). Also, I think it would have been instructive to discuss the asymmetric nature of the reciprocating motion, as its mainly "forward" nature, removing material from root canals, is relevant to the measured outcome of post-treatment pain. The article is otherwise coherent and readable.

Experimental design

I have no major critical comments on the experimental design. It would have been ideal if each patient could have had one RCT done by each method. Obviously, this would have made the study much more difficult. I think the stated lack of a "learning curve" for the Reciproc phase might be considered a minor defect, as the users' facility with the instrument system would reasonably have an influence on the results. Having said that, I recognize that the authors were making an effort to simulate the "heterogeneous conditions of everyday life" in their practices.

Validity of the findings

Since the reciprocating action of the powered files was mostly in the "forward" direction, bringing debris out of the root canal, it is not surprising that post-treatment pain would be on the same order as that observed after hand instrumentation. Discussion of this in comparison to "fully-forward" (non-reciprocating) systems would have been relevant and of interest. Also it does not surprise that operator fatigue is less with a powered system; comparison of reciprocating and non-reciprocating systems would have been of interest. Quite possibly the patients' (relatively minor) complaints about the Reciproc treatments had to do with the vibration associated with the reciprocation.
One might consider the provision of the instruments "free of cost" to be a minor conflict of interest; however, the results do not show evidence of bias.

Additional comments

I hope my remarks are useful. I will be glad to provide the spelling/grammar comments... they are currently not in "digital" format, but I can transcribe them. These are the "minor revisions" I refer to below.
I undertook this review with only a couple weeks' notice at a particularly busy time; I have tried to give a thorough review.

---

## Round 0.2 · accepted · Accept

Thank you for your patience.

·

Basic reporting

OK

Experimental design

OK

Validity of the findings

OK

Additional comments

paper is suitable for publication